# *Weizmannia Coagulans* BC99 Prevents Loperamide-Induced Functional Constipation in Mice Through Increased Intestinal Peristalsis and Modulation of Gut Microbiota Dysbiosis

**DOI:** 10.3390/nu17101729

**Published:** 2025-05-20

**Authors:** Cheng Li, Ying Wu, Hua Liang, Yao Dong, Shuguang Fang, Pan-Young Jeong, Hye-Rim Kim, Shaobin Gu

**Affiliations:** 1College of Food and Bioengineering, Henan University of Science and Technology, Luoyang 471000, China; 2Henan Engineering Research Center of Food Microbiology, Luoyang 450001, China; 3National Demonstration Center for Experimental Food Processing and Safety Education, Luoyang 471000, China; 4Department of Research and Development, Wecare Probiotics Co., Ltd., Suzhou 215200, China; 5Germline Stem Cells and Microenvironment Lab, College of Animal Science and Technology, Nanjing Agricultural University, Nanjing 210095, China; 6Life Science Research Institute, Novarex Co., Ltd., Cheongju 28220, Republic of Korea

**Keywords:** constipation, *Weizmannia coagulans*, gut microbiota, short-chain fatty acids, gastrointestinal regulating peptides

## Abstract

**Objectives:** Constipation is one of the most prevalent gastrointestinal disorders worldwide. Recent studies have demonstrated that probiotics may alleviate constipation by restoring gut microbiota balance. **Methods:** This study investigated the effects of *Weizmannia coagulans* BC99 (formerly *Bacillus coagulans* BC99) on gut microbiota and intestinal function in a loperamide-induced mouse model of functional constipation. BALB/c mice were randomly divided into six groups: control, model, phenolphthalein, BC99-L (2 × 10^7^ CFU/day), BC99-M (2 × 10^8^ CFU/day), and BC99-H (2 × 10^9^ CFU/day). After 14 days of supplementation, constipation was induced in mice via loperamide administration. **Results:** BC99 significantly increased fecal water content, gastrointestinal transit rate, microbial metabolic activity, and butyric acid production, while decreasing the abundance of inflammation-related metabolic pathways. Moreover, BC99 improved levels of gastrointestinal regulatory peptides, including motilin and somatostatin. The Firmicutes-to-Bacteroidetes ratio was elevated in the BC99-M and the BC99-H group compared to the model group, indicating that BC99 effectively modulated gut microbiota composition and host biosynthetic pathways. **Conclusions:**  *Weizmannia coagulans* BC99 alleviated and prevented loperamide-induced functional constipation in mice by enhancing intestinal peristalsis and restoring gut microbial homeostasis.

## 1. Introduction

Chronic constipation is a common functional gastrointestinal disorder, with a global prevalence ranging from 2 to 20% [1,2]. It is typically characterized by infrequent, difficult, or incomplete defecation [3]. The condition is often persistent and recurrent, significantly impairing patients’ physical and mental health, as well as their overall quality of life. Increasing evidence suggests that chronic constipation is frequently associated with gut microbiota dysbiosis [4,5,6,7], which in turn can affect intestinal immune function, motility, and epithelial barrier integrity. Notably, the composition of the intestinal microbiota in constipated individuals differs markedly from that of healthy controls. However, current treatment options for functional constipation remain unsatisfactory. Commonly used laxatives may cause side effects, highlighting the need for safe, effective, and sustainable therapeutic strategies.

Probiotics are defined as living microorganisms and are beneficial to the health of the host when administered in adequate amounts [8,9]. Probiotics can affect the colon peristalsis rate, inhibit the growth of pathogenic bacteria, and promote the production of short-chain fatty acids [10]. The intestinal pH can be modulated by short-chain fatty acids (SCFAs) and lactic acid, potentially exerting beneficial effects on colon peristalsis and constipation relief. Through the stimulation of neural activity and peptide YY secretion, SCFAs have been shown to enhance murine colonic motility. Although there are studies that certain *Lactobacillus* has the effect of improving constipation, such *Lactobacillus* still faces the problem of low colonization rate, which makes it difficult to play the full role of probiotics [11,12]. *Weizmannia coagulans* is a Gram-positive bacteria rod-shaped bacterium with the characteristics of both *Lactobacillus* and *Bacillus*. When the growth conditions of *Weizmannia coagulans* deteriorate, it transforms from a vegetative cell into a stress-resistant spore. Its spore is resistant to acid and bile salts, with 85% of spores able to survive and colonize in the gastrointestinal tract. Based on this *Bacillus* characteristic, *Weizmannia coagulans* is able to maximize the function of inhibiting the growth of harmful bacteria, regulating human intestinal health, and improving digestion and immunity [13,14], this also gives it the potential to relieve constipation.

Given this context, a constipation model was constructed by loperamide induction. Loperamide, an opioid receptor agonist, induces constipation by inhibiting intestinal μ-receptor-mediated peristalsis and reducing intestinal fluid secretion. The model is widely used to simulate the pathological features of functional constipation. It has been shown that loperamide significantly prolongs intestinal transit time and reduces fecal water content, which is highly similar to the human constipation phenotype [15]. The ameliorative effects of the *Weizmannia coagulans* BC99 in mice with induced constipation through monitoring alterations in intestinal microbiota composition, assessing fluctuations in fecal short-chain fatty acid concentrations, and evaluating modulations in regulatory peptide expression.

## 2. Materials and Methods

### 2.1. Animals Experimental Design

Sixty 6-week-old BALB/C male mice (20 ± 2 g) were obtained from the Laboratory Animal Center of Henan University of Science and Technology. Animals were housed under controlled conditions (temperature: 21–25 °C; humidity: 48–55%; 12 h light/dark cycle) with free access to standardized feed and water. Health status was regularly monitored to exclude other variables. All animal procedures were approved by the Ethics Committee of Henan University of Science and Technology (Protocol No. 202312004; approval date: 15 December 2023).

After a one-week acclimatization period, we referred to the Taheri [16] method and made improvements by using a randomized block design grouping method. Based on previous research [17], different doses of probiotics were used for intervention. The mice were randomly divided into six groups: control, model, phenolphthalein (positive control), BC99-L group (2 × 10^7^ CFU/day), BC99-M group (2 × 10^8^ CFU/day), and BC99-H group (2 × 10^9^ CFU/day). The model and treatment groups received their respective interventions for 14 days. From day 15 to day 18, all groups except the control group were administered loperamide (10 mg/kg body weight) to induce constipation. Phenolphthalein was administered at 70 mg/kg for 14 days as a positive control. The experimental timeline is summarized in Figure 1A.

### 2.2. Body Weight

Body weight was measured at the beginning and end of the experiment. Weight gain was calculated as the difference between final and initial body weight.

### 2.3. Determination of Water Content of Defecation

On days 7, 14 and 18, fecal samples were collected between 9:00 and 11:00 a.m. The method was as follows: the mice in each group were placed in a clean cage alone, and the collected stool was put into a 2 mL EP tube. After the collection, the stool was dried to a constant weight under 65 °C conditions. The moisture level of the fecal sample was determined by subtracting its dehydrated mass from the initial hydrated mass.

### 2.4. Time to First Black Stool

Following an 18 h fasting period (with access to water), mice were administered loperamide (or saline in the control group), followed by oral gavage of activated charcoal 1 h later. Mice were then housed individually, and the time from charcoal administration to the appearance of the first black stool was recorded [18,19].

### 2.5. Gastrointestinal Transit Rate

The mice underwent overnight fasting with free access to water. A 0.2 mL suspension of activated charcoal was administered through oral gavage 18 h later. Following a 25 min interval, euthanasia was performed via cervical dislocation. Blood samples were collected into centrifuge tubes and subjected to centrifugation at 3000 revolutions per minute for 6 min to isolate serum. The abdominal cavity was subsequently dissected to expose the mesenteric tissue, with the entire small intestine spanning from the pyloric sphincter to the cecum being meticulously excised. The excised intestinal segment was straightened on a flat surface to ensure accurate measurement. Gastrointestinal motility was quantified by calculating the proportion of the charcoal transit distance relative to the total intestinal length [20,21].

### 2.6. Microbiological Analysis

The microbial diversity in the contents of the mouse cecum was analyzed using 16S rRNA high-throughput sequencing. High-throughput sequencing was performed on the Illumina MiSeq platform. Then, according to the results of OTU analysis, species annotation and diversity analysis were carried out on the samples.

### 2.7. Short-Chain Fatty Acid (SCFA) Analysis

A 0.1 g fecal sample was measured, mixed with 2 mL of distilled water, and subjected to centrifugation at 12,000 rpm for two minutes. After filtering through a 0.22 m filter, 0.20 mL of the supernatant, 0.70 mL of sterile water, and 0.10 mL of 100 ug/m n-butanol were used to prepare the sample for GC detection. The standard solution was formulated by incorporating three SCFAs standards: acetic acid, propionic acid, butyric acid. Next, 0.20 mL of mixed standard solution, 0.70 mL of sterile water, and 0.10 mL of n-butanol were added to the sample, so that the N-butanol sampling concentration was 100 μg/mL, and N-butanol was used as the reference. The average of the peak area of n-butanol was measured several times to correct for the batch-to-batch error. Subsequently, the standard curve could be generated by testing it on the machine, using the concentration of the standard acid as the x-axis and the peak area as the y-axis [22,23].

The analysis was performed using a 6890 N gas chromatograph equipped with a polarity-modified HP-FFAP capillary column (30 m × 0.25 µm × 0.25 µm) and a flame ionization detector (FID). Helium served as the carrier gas with a flow rate of 2 mL/min, while a 10:1 split ratio and 1 µL injection volume were applied. The injector temperature was maintained at 240 °C. The temperature program initiated at 100 °C, followed by a 7.5 °C/min ramp to 140 °C, and then a rapid 60 °C/min increase to 200 °C with a 3 min hold. The FID operated at 220 °C for ionization. Full-spectrum acquisition mode was employed in mass spectrometry. Quantification of SCFAs in µmol/g sample units was achieved through an external calibration curve.

### 2.8. Serum Gastrointestinal Regulatory Peptides Analysis

Motility hormone (motilin) and somatostatin in the serum were analyzed using an enzyme-linked immunosorbent assay (ELISA) following the manufacturer’s instructions. These kits were acquired from BYabscience, Nanjing, China. The results are expressed as μg/mL.

### 2.9. Statistical Analysis

Experimental data are presented as mean ± standard deviation. Between-group comparison: If the data meet the assumptions of normal distribution and homogeneity of variance, independent sample t-tests can be used. If these assumptions are not met, the Mann–Whitney U test can be used. Multiple-group comparison: If the data meet the assumptions of normal distribution and homogeneity of variance, one-way ANOVA analysis can be used. If the data do not meet the assumptions of ANOVA, the Kruskal–Wallis test can be used. The analysis was carried out using SPSS 25.0 at a significance level of 0.05. Origin 2021 and GraphPad Prism 9.5 statistical analysis software were used for analysis.

## 3. Results

### 3.1. Body Weight Change

The results of body weight of experimental animals are shown in Figure 1B. Before loperamide administration, the weight of mice in each group increased continually. During the experimental phase, commencing on day fifteen, a reduction in body mass was observed in both the model group and the intervention group relative to the control group. Notably, the probiotic-treated group exhibited a more gradual decline in weight parameters, had no statistical significance, suggesting therapeutic intervention could potentially mitigate constipation-induced weight loss in murine models.

### 3.2. Water Content of Defecation

The constipated mice had fewer fecal water content and weakened intestinal peristalsis function, generally speaking, compared to control group [24,25]. The results for the fecal water content of experimental animals induced with loperamide are shown in Figure 1C. In addition to the BC99-H group, the water content of defecation in the phenolphthalein group was significantly higher than the model group and control group (*p* < 0.05) on the 14th day. On the 18th day, the water content of defecation both in the control group and the treatment groups decreased to different degrees, whereas there was a significant difference in water content of defecation between the model group and the other groups (*p* < 0.05). Furthermore, the water content of fecal defecation in the phenolphthalein group, the BC99-M group, and the BC99-H group was similar to that in the control group (*p* > 0.05), which indicated that the BC99 could increase the water content of defecation and ease constipation. It is worth noting that the intervention of phenolphthalein tablets will inhibit the absorption of water in the intestines, produce a laxative effect, and cause a higher water content in the stool at 14 days.

### 3.3. The Time of the First Black Stool Defecation

The timing of initial melena occurrence was assessed to determine BC99’s efficacy in alleviating constipation symptoms [26]. The time to the first black stool defecation was significantly prolonged by 33.88% in the model group compared to the control group (*p* < 0.001). Meanwhile, the time of the first black stool defecation was in the phenolphthalein group, the BC99-L group, the BC99-M group, and the BC99-H group was shorter than the model groups, with a significant reduction of 35.91% in the BC99-H group (*p* < 0.001). According to the defecation time, it indicated that the BC99 increased the intestinal movement of mice and relieved constipation effectively (Figure 1D).

### 3.4. Gastrointestinal Peristalsis

The rate of gastrointestinal propulsion is a direct reflection of the small intestine’s movement [27]. Gastrointestinal peristalsis rate was significantly reduced by 20.37% in the model group compared to the control group (*p* < 0.01), indicating the success of loperamide-induced constipation in mice. The rate of gastrointestinal peristalsis of mice in the treatment group was significantly higher than that of the model group, in which the rate of gastrointestinal peristalsis of mice in the BC99-H group was significantly increased. In conclusion, BC99 is effective in mice and could promote gastrointestinal peristalsis and relieve constipation (Figure 1E).

### 3.5. Effects of BC99 on SCFAs Levels

The effects of BC99 on SCFAs levels are depicted in Figure 2. At day 14, there were no statistically significant differences in the levels of propionic acid and butyric acid between the groups. Only acetic acid was significantly elevated by BC99 intervention, with a significant increase of 138.11% in the high-dose group compared to the model group (Figure 2A). At day 18, the levels of acetic acid, propionic acid, and butyric acid were significantly reduced by 56.5%, 79.11%, and 41.21%, respectively, in the model group compared to the control group. After the intervention of BC99, the levels of short-chain fatty acids were significantly restored, in which the levels of acetic acid, propionic acid and butyric acid in the BC99-H group were significantly elevated by 460.98%, 423.9% and 96.87%, respectively, compared with the model group (Figure 2B).

### 3.6. Effects of BC99 on Motilin and Somatostatin in Mice with Constipation

The serum levels of gastrointestinal regulatory peptides in each group are shown in Figure 3. Gastrointestinal regulatory peptide plays a very important role in regulating gastrointestinal peristalsis and the level of gastrointestinal regulatory peptide, which is closely related to the degree of constipation. Motilin plays a role in the upper digestive tract and can promote gastrointestinal peristalsis [28]. Motilin levels were significantly reduced by 32.37% in the model group compared to the control group (*p* < 0.001). Meanwhile, the levels of motilin in the treatment groups were significantly higher than those in the model group (*p* < 0.05). Among them, the motilin level in the BC99-H group was significantly higher than that in the model group by 64.54%, indicating that BC99 could significantly relieve constipation in mice (Figure 3A). As an inhibitory transmitter, the somatostatin can not only inhibit the release of motilin but also reduce the secretion of digestive juice [29]. The levels of somatostatin was different among the groups. Somatostatin level in the model group was 42.9% higher than that in the control group, and the levels of somatostatin in the treatment groups were significantly lower than those in the model group (*p* < 0.05). Among them, the gastric actin level in the BC99-H group was significantly lower than that in the model group by 40.98%. (Figure 3B). The above results indicated that constipation severely affects gastrointestinal hormone levels in mice, and that administration of BC99 can regulate the secretion levels of gastrointestinal excitatory and inhibitory hormones.

### 3.7. Effects of BC99 on Fecal Microbiota

In order to study the effects of different treatment groups on fecal microbiota, we analyzed the changes in fecal microbiota in the model group, control group, phenolphthalein group, BC99-L group, BC99-M group, and BC99-H group in mice. The total number of colonies varied between groups. Compared with the control model group, the total number of colonies was elevated in the BC99-L and BC99-M groups, while it was relatively flat in the BC99-H group (Appendix A). Modifications in the structure of intestinal microbial communities were observed across the model, control, and treatment groups, as outlined in the subsequent analysis (Figure 4). The ratio of Firmicutes to Bacteroidetes was increased in the BC99 group and the phenolphthalein group compared with the model group, indicating that BC99 could improve the intestinal flora. Interestingly, the ratio of Firmicutes to Bacteroidetes was similar in the BC99-M group and the normal group; however, this ratio decreased in the phenolphthalein group was lower than the normal group. After the mice were induced constipation by loperamide, the relative abundance of *Prevotellaceae*_UCG-001, *Bacteroides* and *Prevotella*_9 in the fecal microbiota of mice decreased and that of *Escherichia-Shigella* and *Alistipes* increased. However, the BC99 group increased the abundance of *Lachnospiraceae*_NK4A136, *Bacillus* and *Mucispirillum*, decreased the abundance of *Escherichia-Shigella* and *Alistipes* (Figure 4C–J).

### 3.8. Functional Characteristics of Fecal Microbiota

In order to better understand the function of the fecal microbiota in mice, PICRUSt was used to analyze high-throughput sequencing data based on 16S rRNA. The probiotic treatment groups had more metabolic function compared with model group. Figure 5 shows that the addition of probiotics enhanced microbial metabolism in ascorbate and aldarate metabolism, secondary bile acid biosynthesis, carotenoid biosynthesis, protein digestion and absorption, glycerolipid metabolism, phenylpropanoid biosynthesis, vitamin B6 metabolism, N-Glycan biosynthesis, and Biotin metabolism. However, probiotics decreased the abundance metabolic pathways associated with inflammation: lipopolysaccharide biosynthesis, steroid hormone biosynthesis and arachidonic acid metabolism.

### 3.9. Correlation Analysis Between Vital Gut Genus and the Main Indicators of Constipation

To investigate potential associations between alterations in gut microbial profiles and core constipation parameters, Spearman’s rank correlation analysis was performed to evaluate relationships between predominant microbial taxa and key clinical outcomes (e.g., initial dark stool evacuation latency, small bowel motility index, etc.). This methodological approach aimed to clarify whether structural modifications in intestinal microbiota composition functionally influence constipation pathophysiology (Figure 6). The time to first black stool defecation in constipated mice exhibited a statistically significant positive association with the relative abundance of *Lactobacillus* and *unelassified_f_Prevotellaceae*, while it exhibited a negative association with *Rikenellaceae_RC9 gut_group*, A2, *norank_f_Desulfovibrionaceae*, *unelassified_f_Lachnospiraceae*, *Muribaculum*, *Streptococcus*, and *Roseburia* (*p* < 0.05). Moreover, the positive associations between the relative abundance of A2 (*p* < 0.01), *norank_f _Desulfovibrionaceae* (*p* < 0.05) and *unelassified_f_Lachnospiraceae* (*p* < 0.05), and small intestinal peristalsis ratio was presented. The water content of defecation had a significant positive association with the relative abundance of *Mucispirillum*, *Bucteroides* and A2 (*p* < 0.05) and a negative association with the relative abundance of *Ruminococcus_torques_group* and *Alistipes* (*p* < 0.05). Together, these results indicate that the regulation of BC99 on vital bacterial genera further affects constipation.

## 4. Discussion

Constipation is a prevalent gastrointestinal disorder characterized by infrequent bowel movements (fewer than three per week), hard and dry stools, and incomplete evacuation. Therefore, effective strategies are needed to alleviate its symptoms. Meta-analyses have indicated that probiotics exert protective effects against a range of conditions, including cardiovascular diseases, age-related disorders, and gastrointestinal dysfunctions [30,31,32]. Specifically, probiotic supplementation has been shown to enhance intestinal peristalsis, increase defecation frequency, and improve stool consistency. The purpose of this study was to determine whether BC99 has a relieving effect on constipation. The results suggest that the BC99-H group can effectively prevent or relieve constipation symptoms and decrease the water content of defecation, time of the first black stool defecation, and gastrointestinal peristalsis. Compared with other probiotics (e.g., *Lacticaseibacillus rhamnosus*, *Bifidobacterium animalis*) [33], BC99 was more effective in increasing small intestinal propulsion rate and regulating ss gastric motility levels, which may be related to its sporulation characteristics and colonization ability.

Body weight is commonly used as an indirect marker of general health and gastrointestinal function in murine constipation models [34]. Our study revealed that after gavage of loperamide, the body mass of mice in the model group decreased most severely, which was in agreement with a previous study by Li [35]. In contrast, mice in the BC99-M and BC99-H groups exhibited minimal reduction in fecal water content following loperamide exposure, and their values were not significantly different from the control group, aligning with the results reported by McKenney et al. [36]. The time to the first black stool defecation reflects colonic transit time [37]. These results revealed that the time to the first black stool defecation was the shortest in the group that received a medium dose of BC99 or a high dose of BC99 or phenolphthalein and the longest in the model group. Thus, according to the time of defecation, a medium dose of BC99 and a high dose of BC99 can increase the intestinal movement of mice and effectively relieve constipation. Intestinal peristalsis can promote the movement of the contents of the small intestine. Therefore, fast intestinal peristalsis can be detected by measuring the rate of intestinal propulsion, which is conducive to excretion; otherwise, constipation may occur [38]. In the present study, gastrointestinal peristalsis was the highest in the BC99-H group.

Chronic constipation is frequently accompanied by alterations in gut microbiota. Our results showed that BC99 could improve the ratio of Firmicutes to Bacteroidetes. The intestinal flora is mainly composed of Firmicutes, Bacteroidetes and Actinobacteria. There was less Firmicutes in the intestinal tract of mice in the model group after constipation [39]. In addition, compared to the phylum level, we believe that the analysis at the genus level is more important in gut microbiota analysis. It can further reflect the relationship between the gut microbiota of BC99 and disease. Genus level analysis is consistent with our hypothesis, as the high-dose intervention of BC99 increased the abundance of the genus *Lachnospiraceae* belonging to the Firmicutes phylum, thereby increasing the level of short-chain fatty acids to alleviate constipation. Wintola et al. [40] previously demonstrated that the transplantation of gut microbiota from constipated patients, but not from healthy individuals, induced constipation-like symptoms in antibiotic-depleted mice, underscoring the causal role of dysbiosis.

Microbially produced compounds and SCFAs, which represent hallmarks of intestinal microbial communities and are involved in modulating gut-associated immune responses, serve as critical markers for evaluating therapeutic microbial profiles in inflammatory bowel disease. SCFAs, such as acetic acid, propionic acid, and butyric acid, represent the principal metabolites generated through anaerobic microbial fermentation of indigestible carbohydrates in the colon. These volatile organic compounds are synthesized by gut microbiota and play critical roles in maintaining intestinal homeostasis. Our finding that the concentrations of SCFAs decreased in the model group was consistent with a previous study by Gough et al. [37] who found a decrease in the concentration of SCFAs in the stools of patients with irritable bowel syndrome. The administration of BC99 stimulated higher levels of short-chain fatty acid generation throughout the colon. However, lower-dose probiotic interventions showed limited efficacy in promoting SCFA generation in colonic regions beyond the cecal region. In addition, at 14 days, we noticed that only acetic acid levels increased. Acetic acid is the primary fermentation product of most intestinal anaerobic bacteria, and can also be generated by reductive acetogenesis, while propionic acid and butyric acid are produced by different groups of intestinal bacteria. We speculate that by day 14, BC99 had to some extent already changed the structure of the intestinal microbiota, promoting an increase in the abundance of acetic-acid-producing bacteria in the intestine (e.g., Lachnospiraceae) [41], leading to a more significant change in acetic acid levels. Studies have shown that SCFAs reduced the amplitude of guinea pig terminal and the threshold, and increased the frequency of peristaltic contraction, with guinea pig ileum terminal in vitro as the experimental object [39]. It can be seen from the comprehensive relief of constipation that increasing the contents of acetate, propionate, and butyrate is helpful to regulate bowel movements.

The level of gastrointestinal regulating peptide is closely related to the degree of constipation [39]. Excitatory gastrointestinal regulatory peptide motilin affects the transport of water and electrolytes, promotes gastrointestinal motility, stimulates parietal cells to secrete hydrochloric acid, and stimulates the secretion of pancreatic juice and bile [42]. In this study, the levels of motilin in the model group were lower than those in the control group, whereas levels of motilin were significantly higher in the BC99-H group than those in the model group. This result is supported by Vernocchi et al. [43], who showed that the secretion of motilin in patients with irritable bowel syndrome was lower than that in healthy normal. In addition, Mortensen et al. [44] reported that the levels of motilin in patients with diarrhea were higher than those in patients with constipation. Somatostatin is the most widely distributed inhibitory hormone, secreted by enteroendocrine D cells, and its main role is to inhibit the release of various excitatory brain–gut peptides, inhibit gastrointestinal motility, and slow down intestinal transit time [45]. Somatostatin can inhibit the release of motilin. The results of this study showed that the levels of somatostatin in the model group were higher than those in the control group, whereas these levels were significantly lower in the BC99-H group than those the control group. Therefore, this observation demonstrated that reduced motilin levels and elevated somatostatin activity could serve as potential contributors to constipation. However, understanding the specific mechanism of how BC99 regulated the levels of gastrointestinal regulatory peptides is a limitation of this study. Correlation findings showed a significant relationship between gastrointestinal regulatory peptides and intestinal flora, indicating an association between the two. BC99 may stimulate intestinal mucosal cells via SCFAs (e.g., butyric acid), which regulate gastrointestinal motility on the one hand by stimulating the secretion of gastrointestinal regulatory peptides from intestinal L cells through the activation of GPR41 and GPR43 receptors [46]. On the other hand, SCFAs stimulate the release of 5-hydroxytryptamine (5-HT) from enteroendocrine cells, which regulates gastrointestinal peptide levels and activates vagal signaling to accelerate intestinal contractions [47].

Functional prediction analysis further suggested that BC99 enhanced microbial metabolic functions while suppressing inflammation-related pathways. Reigstad et al. [48] reported that the presence of endogenous spore-forming bacteria in the gut increased colonic motility by modulating microbial metabolism. In our study, BC99 significantly enriched beneficial microbial taxa involved in SCFAs and bile acid synthesis, including *Mucispirillum*, *Bacteroides*, *Oscillibacter*, *Rikenellaceae_RC9_gut_group*, *norank_f_Desulfovibrionaceae*, *unclassified_f_Lachnospiraceae*, *Muribaculum*, and *Streptococcus* [49,50]. The abundance of genus A2 was directly correlated with SCFAs levels and is known to be involved in multiple gastrointestinal metabolic pathways.

## 5. Conclusions

In conclusion, this study provides preliminary evidence that BC99 can adjust the structure of gut microbiota; increase the abundance of *Mucispirillum*, *Bacteroides*, *Oscillibacter,* and other genera; and promote the generation of short chain fatty acids to promote gastrointestinal motility. On the other hand, BC99 also regulates the levels of gastrointestinal regulatory peptides, thereby stimulating gastrointestinal peristalsis and ultimately relieving loperamide-induced functional constipation disorders in mice.

We believe that this article still has some limitations. For example, the specific molecular mechanism by which BC99 regulates gastrointestinal peptides has not been clarified, and we hypothesize that it may be related to SCFAs, but further experimental verification is needed. Second, 16S rRNA sequencing revealed structural changes in the colony, but the lack of metabolomic data made it difficult to fully resolve the regulatory mechanisms of functional metabolic pathways by pathway prediction alone.

## Figures and Tables

**Figure 1 nutrients-17-01729-f001:**
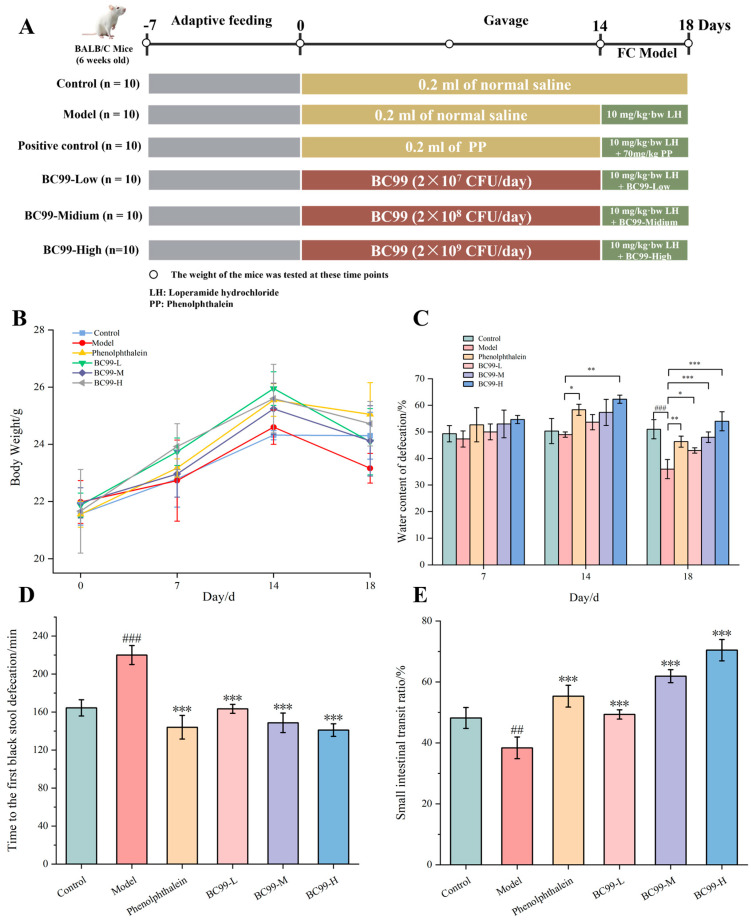
(**A**) The flowchart of the study design. (**B**) Changes in body weight of mice in each group. (**C**) Water content of defecation of mice. (**D**) First black stool defecation time of mice in each group. (**E**) The small intestinal transit rate of mice in each group. Mean values with different letters over the bars are significantly different according to Duncan’s multiple range test. ## *p* < 0.01, ### *p* < 0.001 vs. the control group. * *p* < 0.05, ** *p* < 0.01, *** *p* < 0.001 vs. the Model group.

**Figure 2 nutrients-17-01729-f002:**
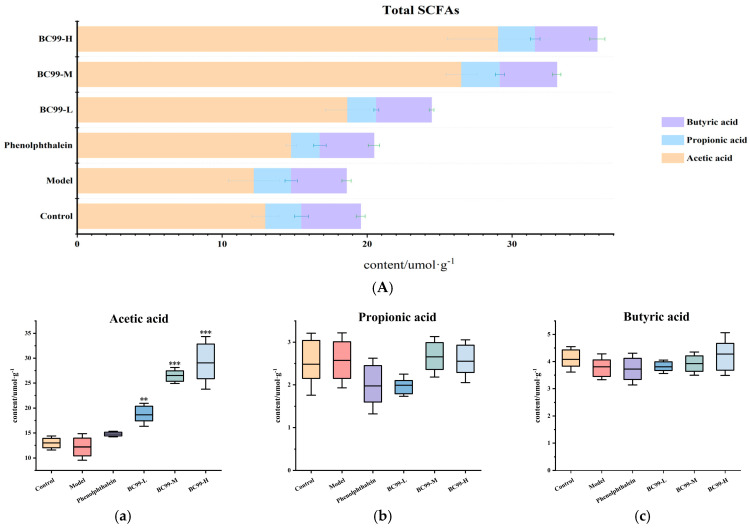
Short-chain fatty acids (SCFAs) of feces. (**A**) SCFAs of feces before constipation. (**B**) SCFAs of feces after constipation. (**a**) Acetic acid content. (**b**) Propionic acid content. (**c**) Butyric acid content. ## *p* < 0.01, ### *p* < 0.001 vs. the control group. ** *p* < 0.01, *** *p* < 0.001 vs. the model group.

**Figure 3 nutrients-17-01729-f003:**
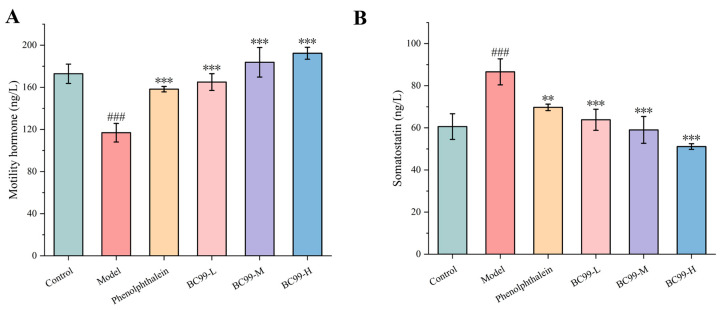
The changes in motilin and somatostatin in different treatment groups. (**A**) Motilin content; (**B**) somatostatin content. ### *p* < 0.001 vs. the control group. ** *p* < 0.01, *** *p* < 0.001 vs. the model group.

**Figure 4 nutrients-17-01729-f004:**
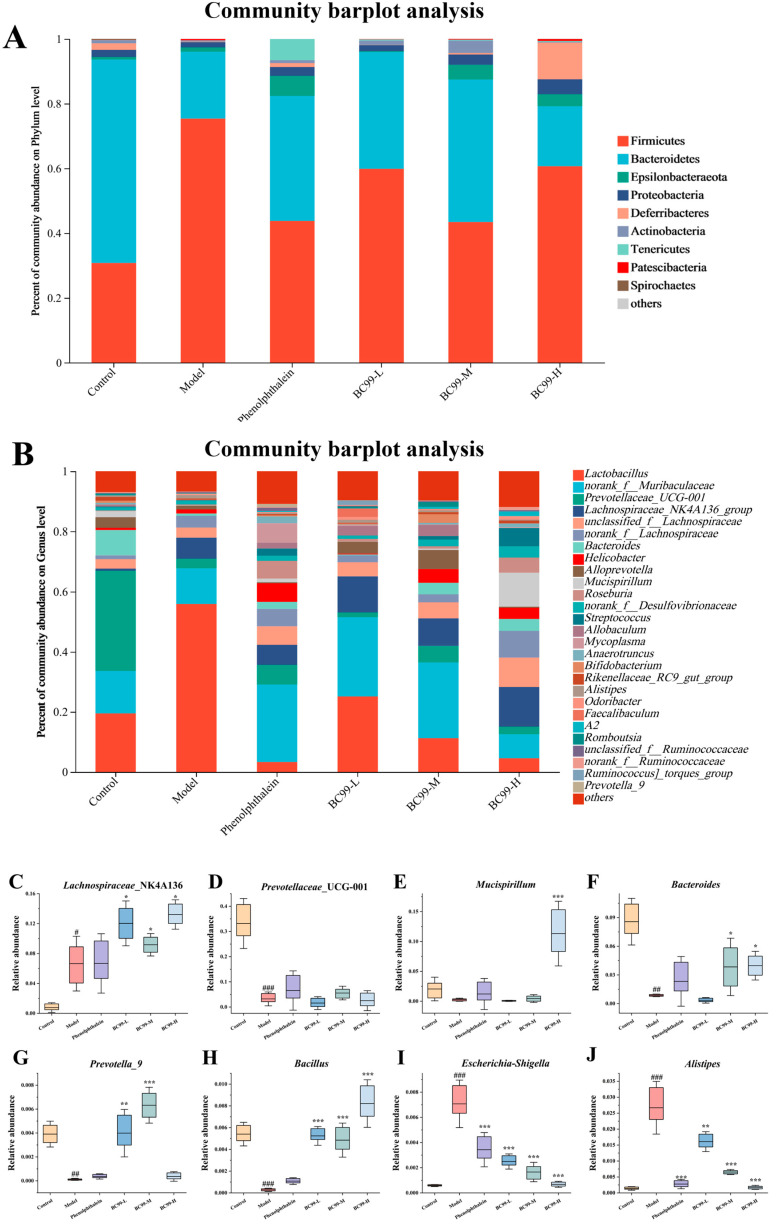
Bacterial similarity and species richness estimates of bacterial 16S rRNA gene sequences obtained by PCR amplification with 95% sequence similarity. (**A**) Relative abundance of main phyla in different groups. (**B**) Relative abundance of main genera in different groups. (**C**) Relative abundance of *Lachnospiraceae*_NK4A136. (**D**) Relative abundance of *Prevotellaceqe UCG-001*. (**E**) Relative abundance of *Mucispirillum*. (**F**) Relative abundance of *Bacteroides*. (**G**) Relative abundance of *Prevotella 9*. (**H**) Relative abundance of *Bacillus*. (**I**) Relative abundance of *Escherichia*-*Shigella*. (**J**) Relative abundance of *Alistipes*. # *p* < 0.05, ## *p* < 0.01, ### *p* < 0.001 vs. the control group. * *p* < 0.05, ** *p* < 0.01, *** *p* < 0.001 vs. the model group.

**Figure 5 nutrients-17-01729-f005:**
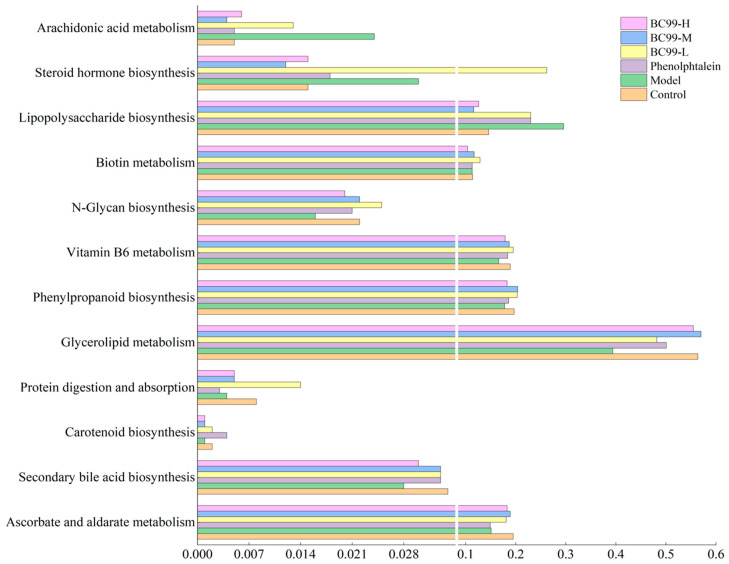
Effect of BC99 on the KEGG pathway of intestinal flora in constipated mice based on PICRUST prediction.

**Figure 6 nutrients-17-01729-f006:**
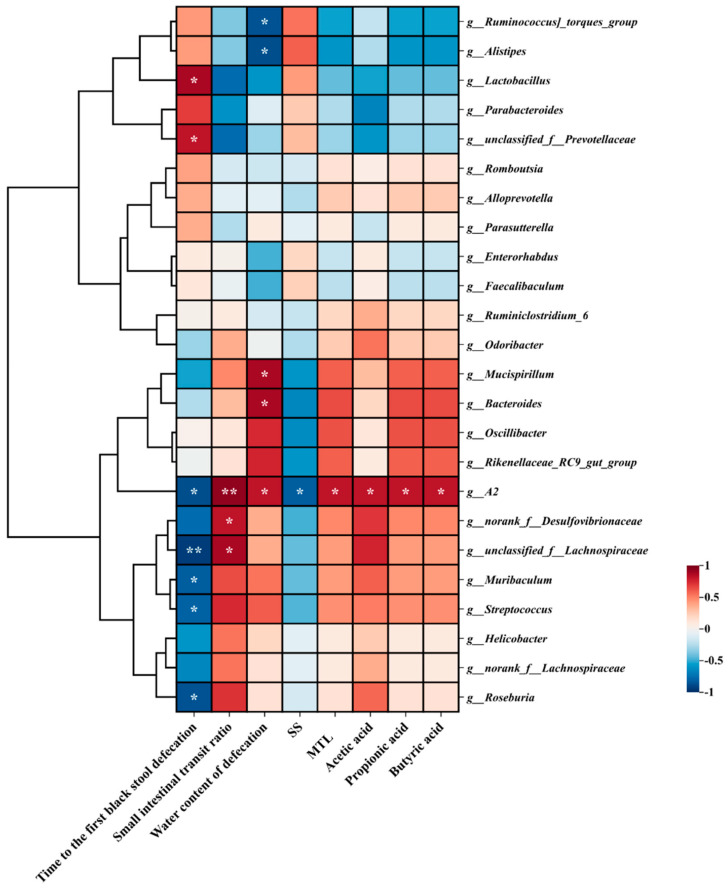
The Spearman correlation between the dominant genera of intestinal microbiota and time to first black stool defecation, small intestinal transit ratio, water content of defecation, SOMATOSTATIN, MOTILIN, and SCFAs. The R values are represented by gradient colors, where red and blue cells indicate positive and negative correlations, respectively; * *p* < 0.05, ** *p* < 0.01.

## Data Availability

The data presented in this study are available on request from the corresponding author. The 16S rRNA gene sequence of mice fecal has been submitted to NCBI genome database under PRJNA1097236.

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
