# Peer review of "Weizmannia Coagulans BC99 Prevents Loperamide-Induced Functional Constipation in Mice Through Increased Intestinal Peristalsis and Modulation of Gut Microbiota Dysbiosis"

_nutrients, 2025, doi:10.3390/nu17101729_

Round 1
Reviewer 1 Report
Comments and Suggestions for Authors
Comments to Manuscript (Cheng Li et al., Nutrients, ID-3584364) “Weizmannella coagulans BC99 prevents loperamide-induced functional constipation in mice through increased intestinal peristalsis and modulation of gut microbiota dysbiosis.”
Li C. et al demonstrate that oral administration of probiotic bacteria Weizmannella coagulans BC99 significantly improved functional constipation in loperamide-treated mice. The authors found that pretreatment of mice with oral administration of BC99 improved loperamide-induced constipation. BC99 dose-dependently increased intestinal peristalsis and fecal water content. The authors also analysed gut microbiota to investigate the ameliorative mechanisms of this probiotic bacteria. This article contains attractive topics for many readers in the related fields, but revisions described below are necessary for publication.
Major comments
- In Introduction section, please add some explanation about loperamide-induced constipation model with citing references.
- Is the phenolphthaleine (PP) used as blocking agent against loperamide? The PP treatment in positive control group (Fig 1A flowchart) is different from the explanation in Materials and Methods (Animal experimental design, line 7).
- How about were the total bacterial count in cecum of model mice with/without BC99 treatment? I think it important to compare (or discuss) the amounts of intestinal bacteria among the experimental groups because the fecal SCFAs dramatically increased in BC99-treated groups. The authors suggest the contribution of increased abundance of Lachnospiraceae to acetic acid production. Could the BC99 increase total amount of microbiota to produce more SCFAs?
- Figure 5 showed that BC99 enhanced microbial metabolisms such as secondary bile acid biosysnthesis. Is it possible that secondary bile acids or other metabolites involve in BC99-mediated amelioration of constipation?
Minor comments
- The name of bacteria, phylum, family, genus should be correctly written. Please check again throughout the manuscript.
- Some figure panels are too small and hard to read.
- Page 7, line 13-14, This sentence explain the role of somatostatin, so please cite references.
- Page 4, line 42, 43, “ss” may be deleted. (assay?, expressed?)
- Page 4, line 21-25, Microbiological analysis, are the microbiota of cecum contents analysed by 16S targeting metagenome sequencing?
- Page 11, line 43, “[33, 34]”, are these references necessary here?

As described in my comments to the authors
Reviewer 2 Report
Comments and Suggestions for Authors
Major:
- Authors should more clearly articulate the rationale for selecting Weizmannia coagulans BC99, including how it differs from other probiotics previously studied in the context of constipation.
- Authors should improve the description of the SCFA analysis protocol by adding details about calibration curves, internal standards, and analytical validation.
- Authors should report exact p-values and specify statistical tests used directly in the figure legends.
- Authors should provide a scientific rationale for the selected dosages of BC99. Were these doses based on previous literature, preliminary data, or regulatory recommendations?
- Authors should discuss whether the study controlled for confounding variables such as food and water intake, stress levels, or cage conditions that might influence intestinal motility and microbiota composition.
- Authors should elaborate on the potential mechanisms BC99 influences gastrointestinal regulatory peptides such as motilin and somatostatin. Is this a direct microbial effect or mediated via metabolites like SCFAs?
- The authors should consider adding a conclusion section that summarises all the results.
Minor:
- The manuscript would benefit from a thorough language revision to correct grammatical errors and awkward phrasing.
- Figure legends should define all group labels and statistical notations.
Minor linguistic and stylistic corrections are required.
Reviewer 3 Report
Comments and Suggestions for Authors
The title is informative but too long, may be the authors consider shortening it by removing redundant phrasing
The abstract contains several grammar errors and awkward phrasing – please check.
For example:
“After 14 days supplementation, the mice were induced constipation by loperamide.”
should be “After 14 days of supplementation, constipation was induced in the mice using loperamide.”
The introduction provides a good background, but is redundant in several areas, please focus the introduction more narrowly on Weizmannia coagulans and its novelty compared to other probiotics in this specific context.
The randomization and blinding procedures are not described. Without these, results may be biased – please add clear details about ethical approvals (which is later repeated in the discussion — move it here
ELISA method is referred to vaguely, please specify ELISA kit brands
Most results are described as “significant” without showing actual numbers. Authors must include mean ± SD and exact p-values for all comparisons. Example: intestinal transit rates, SCFA levels, and motilin/somatostatin concentrations.
There are some contradictions in SCFA results, in one part the authors says there’s no significant difference in butyric/propionic acid (see Figure 2), then later claims a significant increase “The content of propionate acid in the BC99-H group was significantly higher than that in other treatment groups (p<0.05). Additionally, the content of butyric acid in the BC99-M group and the BC99-H group was significantly higher than that in other treatment groups (p<0.05).” This needs to be clarified and backed with numbers.
The discussion restates almost every result instead of explaining why those effects might happen. Example: motilin and somatostatin are mentioned again with no mechanistic link to microbiota or SCFAs.
Some claims are too strong for the data shown, for example phrases like “BC99 significantly relieved constipation by regulating hormone secretion” suggest causality that hasn’t been proven
There’s limited comparison with other probiotic studies, in my opinion authors should discuss how BC99 performs versus other strains like L. casei, B. lactis, or other Bacillus strains already tested in similar models.
The conclusions sound too confident for a mouse study. It should be made clear that these are preliminary results. Also, there’s no mention of limitations — that’s important and should be added.
Comments on the Quality of English LanguageThe abstract contains several grammar errors and awkward phrasing – please check.
For example:
“After 14 days supplementation, the mice were induced constipation by loperamide.”
should be “After 14 days of supplementation, constipation was induced in the mice using loperamide.”
Overuse of passive voice and inconsistent tense makes the paper difficult to follow. A native English-speaking should review the manuscript.
Round 2
Reviewer 2 Report
Comments and Suggestions for Authors
The authors have thoroughly addressed all my comments. I think the only changes needed to the manuscript are figures 2a, b and c, and 4c-j; they require enlargement to increase readability, but do not provide a clear possibility of reading the results.
In addition, citation 11 does not have a DOI number; if it is a book fragment or another source, additional information should be provided to find the source material.
Reviewer 3 Report
Comments and Suggestions for Authors
The authors have made the suggested revisions based on the reviewers' comments and have improved the manuscript accordingly.